# Enhanced Heterologous Production of Glycosyltransferase UGT76G1 by Co-Expression of Endogenous *prpD* and *malK* in *Escherichia coli* and Its Transglycosylation Application in Production of Rebaudioside

**DOI:** 10.3390/ijms21165752

**Published:** 2020-08-11

**Authors:** Wenju Shu, Hongchen Zheng, Xiaoping Fu, Jie Zhen, Ming Tan, Jianyong Xu, Xingya Zhao, Shibin Yang, Hui Song, Yanhe Ma

**Affiliations:** 1University of Chinese Academy of Sciences, Beijing 100049, China; shuwenju@tib.cas.cn (W.S.); zhao_xy@tib.cas.cn (X.Z.); yangshb@tib.cas.cn (S.Y.); 2Industrial Enzymes National Engineering Laboratory, Tianjin Institute of Industrial Biotechnology, Chinese Academy of Sciences, Tianjin 300308, China; fu_xp@tib.cas.cn (X.F.); zhen_j@tib.cas.cn (J.Z.); tan_m@tib.cas.cn (M.T.); xu_jy@tib.cas.cn (J.X.); 3Tianjin Key Laboratory for Industrial Biological Systems and Bioprocessing Engineering, Tianjin Institute of Industrial Biotechnology, Chinese Academy of Sciences, Tianjin 300308, China

**Keywords:** steviol glycosides, fusion partner, efficient *E. coli* expression system, *prpD*, *malK*, co-expression, uridine diphosphate dependent glucosyltransferases (UGTs), enzymatic biotransformation

## Abstract

Steviol glycosides (SGs) with zero calories and high-intensity sweetness are the best substitutes of sugar for the human diet. Uridine diphosphate dependent glycosyltransferase (UGT) UGT76G1, as a key enzyme for the biosynthesis of SGs with a low heterologous expression level, hinders its application. In this study, a suitable fusion partner, Smt3, was found to enhance the soluble expression of UGT76G1 by 60%. Additionally, a novel strategy to improve the expression of Smt3-UGT76G1 was performed, which co-expressed endogenous genes *prpD* and *malK* in *Escherichia coli*. Notably, this is the first report of constructing an efficient *E. coli* expression system by regulating *prpD* and *malK* expression, which remarkably improved the expression of Smt3-UGT76G1 by 200% as a consequence. Using the high-expression strain *E. coli* BL21 (DE3) M/P-3-S32U produced 1.97 g/L of Smt3-UGT76G1 with a yield rate of 61.6 mg/L/h by fed-batch fermentation in a 10 L fermenter. The final yield of rebadioside A (Reb A) and rebadioside M (Reb M) reached 4.8 g/L and 1.8 g/L, respectively, when catalyzed by Smt3-UGT76G1 in the practical UDP-glucose regeneration transformation system in vitro. This study not only carried out low-cost biotransformation of SGs but also provided a novel strategy for improving expression of heterologous proteins in *E. coli*.

## 1. Introduction

Nowadays, more and more countries are facing a serious public health burden from the consumption of high-calorie sugars such as glucose, fructose, and sucrose [1,2,3]. Diets high in such high-calorie sugar are recognized to cause a serious health problem such as obesity, diabetes, hypertension and cardiac blockage [4,5]. Some countries have levied sugar taxes to reduce sugar consumption [1]. However, a more promising strategy for this problem is to exploit alternative sweeteners with low- or zero-calorie [1]. At present, artificial sweeteners such as aspartame and sucralose are widely used but have not displaced sugar because of their unfavorable effect on health including glucose intolerance and failure to cause weight reduction [6]. Steviol glycosides (SGs) as natural zero-calorie sweeteners with desirable high-intensity sweetness are recognized as the attractive sugar substitutes [7,8].

SGs commonly extracted from stevia plants especially from the leaves of *Stevia rebaudiana* Bertoni. SGs consists of a common diterpenoid steviol backbone and a variable glycone composed mainly of glucose molecules linked by β-glycosidic bonds to the steviol aglycone at the C13-hydroxyl and/or C19-carboxylate [1]. To date, more and more different SGs are identified in stevia plants, among which stevioside (St) and rebadioside A (Reb A) constantly account for the largest proportion [9]. However, the bitter aftertaste of St and Reb A restricts SGs utilization for human consumption and limits their application in the food industry. Reb D (penta-glycoside) and Reb M (hexa-glycoside) which are trace components in stevia plants, but their sweetness potency is up to 350 times greater than that of sucrose and notably have no bitter aftertaste, are recognized to be high-quality sweeteners and the best substitute for sugar [10]. While the production of Reb D and Reb M could not rely on extracting from stevia plants due to their less than 0.1% dry weight content. The enzymatic conversion from St or Reb A by various glycosyltransferases should be a promising strategy.

To date, substantial efforts have been made to reveal glucosylation enzymes and successfully excavated more and more uridine diphosphate (UDP)-dependent glucosyltransferases (UGTs) [11,12,13,14]. The UGTs with high regioselectivity and substrate specificity have attracted remarkable attention due to their great potential application in biotechnology [15,16,17]. In the metabolic glycosylation grid of *Stevia rebaudiana* Bertoni, glucosyltransferase UGT85C2 transfers glucose to the C13-hydroxyl position of the steviol backbone, forming a β-d-glucoside, whereas UGT74G1 is responsible for glucosylating the C19-carboxylic acid-activated group to form an ester. Subsequently, UGT91D2 and UGT76G1 directly catalyze the formation of 1, 2-β-d- and 1, 3-β-d-glucoside bonds at the C13- and C19- positions, respectively [18]. With these four enzymes, the biosynthesis of almost all SGs including St, Reb A, Reb D and Reb M, could be identified [19]. Based on previous reports, the content levels of the main SGs in stevia are presumably markedly influenced by the UGT76G1 which is the key enzyme for the yield of Reb A and Reb M [20]. Furthermore, the *E. coli* expression system has ever been proven as a reliable approach for the heterologous expression of glycosyltransferases [21,22]. But, low catalytic activity and difficult soluble heterologous expression remarkably hinders their practical applications [22]. Here, a series of strategies were carried out to improve the soluble expression of UGT76G1 in *Escherichia coli*. And it is the first report that through coordinately regulated the overexpression of endogenous *prpD* and *malK* to remarkably enhanced the expression of recombinant proteins in a novel *E. coli* expression system. Moreover, recombinant glycosyltransferase produced by the engineered strain constructed in this work has proven to be efficient in high yield of Reb A and Reb M in vitro enzymatic biotransformation.

## 2. Results and Discussion

### 2.1. Enhanced Soluble Expression of UGT76G1 in E. coli BL21 (DE3) by Fusion Partners

First of all, three plasmids pET32a (+), pET22b (+), and pET26b (+) have been used for the expression of UGT76G1. Through SDS-PAGE analysis results, we found that the soluble expression of UGT76G1 was relatively higher with pET32a-UGT76G1 or pET22b-UGT76G1 than that with plasmid pET26b-UGT76G1 (Appendix A). Besides, it showed that recombinant enzymes expressed in different plasmids result in different conversion rates from St to Reb A (Appendix A). The soluble recombinant enzyme expressed by plasmid pET32a-UGT76G1 showed the highest activity which could completely convert St to Reb A in 12 h (Appendix A). While the correspondent conversion rates by the recombinant UGT76G1s from *E. coil* BL21 (pET22b-UGT76G1) and *E. coil* BL21 (pET26b-UGT76G1) were 70% and 9.8% respectively. It is probably because the TrxA fusion partner of pET32a in the N terminal of the recombinant enzyme (TrxA-UGT76G1) enhanced its soluble expression. So the conversion rate from St to Reb A by TrxA-UGT76G1 showed at least 30% higher than that by UGT76G1 (pET22b-UGT76G1) with the same volume dosage of the crude extract enzymes. And previous research also concluded that fusion partners could enhance the soluble expression of heterologous enzymes in *E. coli* cells [21,23,24]. Thus, we selected other five fusion partners (Fh8, MBP, Smt3, DsbA and DsbC) to find a more suitable fusion partner for the soluble expression of UGT76G1 (Figure 1A). The pET32a-UGT76G1 and pET40b-UGT76G1 were used as controls. According to the sodium dodecyl sulfate polyacrylamide gel electrophoresis (SDS-PAGE) analysis, the fusion partner TrxA (pET32a) and Smt3 made remarkable higher soluble expression of recombinant UGT76G1 (Figure 1B). Similarly, the catalytic activities of the recombinant fusion enzymes TrxA-UGT76G1 and Smt3-UGT76G1 showed relatively higher expressions than those of others (Figure 1C). The highest conversion rate from St to Reb A was 30% by the fusion enzyme Smt3-UGT76G1 in 3 h which is 20% higher than that by TrxA-UGT76G1 (Figure 1C). Besides, pET-DsbC-32 and pET40b have the same fusion partner (DsbC) and both produced the same fusion enzymes DsbC-UGT76G1. While the activity of the fusion enzymes expressed with pET-DsbC-32-UGT76G1 showed two-times higher than that with pET40b-UGT76G1 (Figure 1C). It indicated that the plasmid skeleton of pET32a is more suitable for screening the fusion partners to get higher expression of UGT76G1.

### 2.2. Overexpression of UGT76G1 by Co-Expression with prpD and malK

In our previous work, we firstly found that the enhanced expression of endogenous gene *prpD* or *malK* (ethanol regulated) could obviously improve the expression of heterologous catalase [25,26]. The *malK* and *prpD* gene regulates the maltose ATP-binding cassette (ABC) transporter and propanoate metabolism respectively to enhance the carbon metabolism and energy metabolism (TCA cycle) of *E. coli* cells to improve the synthesis of heterologous proteins [26]. However, the effects by coordinately regulated the overexpression of endogenous *prpD* and *malK* remain further to reveal and whether this strategy could be valid for the heterologous expression of plant genes is also not clear. In order to verify the actual effects on the expression of Smt-UGT76G1 by coordinately regulating the overexpression of endogenous *prpD* and *malK*, we constructed a novel *E. coli* overexpression system (*E. coli* BL21 (DE3) M/P-(1–6)) which contains six strains harbouring recombinant plasmids pACYC-184-*prpD*, pACYC-184-*malK*, pACYC-Duet1-*prpD^I^*-*malK^I^*, pACYC-Duet2-*prpD^C^*-*malK^I^*, pACYC-Duet3-*prpD^I^*-*malK^C^*, and pACYC-Duet4-*prpD^C^*-*malK^C^*, respectively in this work (Figure 2 and Table 1). And each of the six recombinant expression strains was used to express the fusion recombinant UGT76G1 through plasmid pET-Smt3-32-UGT76G1. The relative soluble expressions (Figure 3A) corresponded to the relative conversion rates from St to Reb A (Figure 3B) by the six recombinant strains. All of the co-expression of *prpD* or/and *malK* with Smt3-UGT76G1 could improve the conversion rate of St compared with the individual expression of Smt3-UGT76G1 (Figure 3B). Among them, the recombinant enzyme Smt3-UGT76G1 produced by strain M/P-3-S32U could completely convert St to Reb A in 6 h which showed the highest activity (Figure 3B). And the recombinant Smt3-UGT76G1 produced by strain M/P-5-S32U also showed high activity with slightly lower than that of strain M/P-3-S32U (Figure 3B). The results of SDS-PAGE analysis also showed the enhanced soluble expression of Smt3-UGT76G1 by co-expression of *prpD* or/and *malK*, especially by co-expression of both *prpD* and *malK* (Figure 3A). According to the conversion rates from St to Reb A by Smt3-UGT76G1 enzymes from different expression strains, the transglycosylation activity of the recombinant Smt3-UGT76G1 in strain M/P-3-S32U showed ≥200% increased than that of Smt3-UGT76G1 in strain S32U. And compared with strain S32U, strain M/P-4-S32U and strain M/P-5-S32U produced 83% and 126% higher transglycosylation activity, respectively. It indicated that coordinately regulating the overexpression of *prpD* and *malK* made the higher enhancement of expression of the recombinant Smt3-UGT76G1. Besides, for soluble expression of Smt3-UGT76G1, the enhanced expression of both *prpD* and *malK* synchronized with the target protein is the best choice. When constitutively overexpressed both *prpD* and *malK*, the growth of strain M/P-6-S32U showed remarkable lower than other strains which resulted in relatively lower expression of Smt3-UGT76G1 (Appendix A and Figure 3). However, we assured that overexpression of endogenous gene *prpD* and *malK* simultaneously indeed could remarkably enhance the expression of heterologous Smt3-UGT76G1 at least in this work. And this is the first report to explore the combinatorial co-expression strategies of *prpD* and *malK* to enhance the expression of the heterologous protein.

### 2.3. Scale-Up Production of Glycosyltransferase Smt3-UGT76G1 by Fed-Batch Fermentation in a 10 L Fermenter

We have successfully constructed an efficient expression strain (M/P-3-S32U) for the soluble production of the recombinant glycosyltransferase Smt3-UGT76G1. Then we respectively used the recombinant strains M/P-3-S32U and S32U for scaling up fed-batch fermentation in a 10 L fermenter. The growth profiles showed that both of the strains reached the highest density at 28–32 h (Figure 4A,B). Compared with strain S32U, the fermentation density of strain M/P-3-S32U was a little higher (Figure 4A,B). However, the protein expression of the recombinant enzyme Smt3-UGT76G1 was remarkably higher by strain M/P-3-S32U than strain S32U (Figure 4C,D). After 32 h fermentation, strain M/P-3-S32U produced the highest soluble amount (1.97 g/L) of recombinant glycosyltransferase Smt3-UGT76G1 (Figure 4C). In that case, the yield rate of Smt3-UGT76G1 was 61.6 mg/L/h. It indicated that the efficient expression strain M/P-3-S32U showed prospective for industrial production of glycosyltransferase.

### 2.4. Biotransformation of SGs by the Recombinant UGT76G1 in Vitro Enzymatic Catalysis

As previously reported, UGT76G1 catalyzes the transfer of a glucose moiety in a β (1–3) manner to an existing glucose residue (Figure 5) at both the C13 and C19 positions of the steviol aglycone [1]. It means that UGT76G1 could catalyze the biotransformation from St and Reb D to Reb A and Reb M, respectively. Moreover, in order to qualify for the practical application value of the recombinant glycosyltranferase Smt3-UGT76G1, we used fusion enzyme Smt3-UGT76G1 and its tag deleted enzyme UGT76G1 (Appendix A) respectively to catalyze both reactions in vitro using a UDP-glucose regeneration reaction system. In the biotransformation from St to Reb A, both Smt3-UGT76G1 and UGT76G1 showed very high catalysis activity which could completely transfer St to Reb A within 2 h (Figure 5B,C). However, during 12 h biotransformation, Smt3-UGT76G1 and UGT76G1 transferred Reb D to Reb M with conversion rates of 70.3% and 72.2% respectively (Figure 5E,F). According to these results, it could be concluded that the fusion enzyme Smt3-UGT76G1 has the similar catalysis activity with its tag deleted enzyme UGT76G1 (Figure 5). In other words, the fusion partner Smt3 did not affect the catalytic activity of UGT76G1 which could skip the tag removal step in the practical applications. Moreover, the velocity of substrate consumption and product production showed that the rate of the glycosylation to C19 is slower than that to the C13 by Smt3-UGT76G1 (Figure 5). Consistently with previous reports [1,18,20]. The final yield of Reb A and Reb M was reached 4.8 g/L and 1.8 g/L in this practical UDP-glucose regeneration transformation system in vitro. It indicated that the recombinant glycosyltransferase Smt3-UGT76G1 is a promising candidate for the industrial production of SGs.

## 3. Materials and Methods

### 3.1. Gene Cloning and Plasmids Construction for Fusion Expression

The codon-optimized gene (UGT76G1) derived from *Stevia rebaudiana* UGT76G1 mRNA (AY345974) was synthesized by General Biosystems Co. Ktd (Anhui, China) and then cloned into pET-32a (+)/pET-40b (+) directly connected to the enterokinase (EK) cleavage site following the TrxA/DsbC fusion partner gene of plasmid. Encoding gene for *Arabidopsis thaliana* sucrose synthase SUS1 (GenBank accession number: 832206), with codons optimized for *E. coli* heterologous expression, was synthesized and cloned into the *E. coli* expression vector pET22b. The resultant recombinant plasmids called pET32a-*UGT76G1* and pET40b-*UGT76G1* and pET22b-*SUS1*, respectively. Gene (Appendix A) encoding each of the five solubility-enhancer proteins Fh8, MBP, Smt3, DsbA and DsbC was separately cloned into pET-32a (+) substituting TrxA in situ (Figure 1A). The resultant recombinant plasmids were named pET-*Fh8*-32, pET-*MBP*-32, pET-*Smt3*-32, pET-*DsbA*-32, and pET-*DsbC*-32, respectively. When the gene *UGT76G1* was cloned into these plasmids, they were called pET-*Fh8*-32-*UGT76G1*, pET-*MBP*-32-*UGT76G1*, pET-*Smt3*-32-*UGT76G1*, pET-*DsbA*-32-*UGT76G1*, and pET-*DsbC*-32-*UGT76G1*, respectively. The gene *UGT76G1* was also cloned into the multiple cloning sites of pET-22b (+) and pET-26b (+) respectively for expression. The recombinant plasmids were constructed using seamless cloning kit (GenScript Co., Ltd., Nanjing, China) with PCR amplification with the appropriate primers (Appendix A). The *E. coli* strain BL21 (DE3) was used as the host for protein expression.

### 3.2. Construction of E. coli Overexpression System by Co-Expressing the Endogenous Genes prpD and malK

Endogenous genes *prpD* and *malK* were cloned using primers (Appendix A) and the genomic DNA of *E. coli* BL21 (DE3) as a template. Then genes *prpD* and *malK* were separately cloned into the plasmid pACYC-184 following the constitutive tet promoter. The resultant recombinant plasmids were named as pACYC-184-*prpD* and pACYC-184-*malK*, respectively (Figure 2). Genes *prpD* and *malK* were separately cloned into the two multiple cloning sites (MCSs) of plasmid pACYC-Duet1 to get a recombinant plasmid as pACYC-Duet1-*prpD^I^*-*malK^I^* (Figure 2). The two MCSs of plasmid pACYC-Duet1 have the same inducible promoter which consist of T7 promoter and lac operator (Figure 2). The first inducible promoter of pACYC-Duet1-*prpD^I^*-*malK^I^* was substituted by the tet promoter from pACYC-184 to form pACYC-Duet2-*prpD^C^*-*malK^I^* (Figure 2). Similarly, the second inducible promoter of pACYC-Duet1-*prpD^I^*-*malK^I^* was displaced by the tet promoter from pACYC-184 to generate pACYC-Duet3-*prpD^I^*-*malK^C^* (Figure 2). When both of the two inducible promoters were replaced by the tet promoters from pACYC-184, the resultant recombinant plasmid was named as pACYC-Duet4-*prpD^C^*-*malK^C^* (Figure 2). Transformation of the *E. coli* BL21 (DE3) with the six recombinant plasmids pACYC-184-*prpD*, pACYC-184-*malK*, pACYC-Duet1-*prpD^I^*-*malK^I^*, pACYC-Duet2-*prpD^C^*-*malK^I^*, pACYC-Duet3-*prpD^I^*-*malK^C^*, and pACYC-Duet4-*prpD^C^*-*malK^C^* lead to the formation of a novel efficient expression system *E. coli* BL21 (DE3) M/P-(1–6) listed in Table 1. And then transformed the plasmid pET-*Smt3*-32-*UGT76G1* to the recombinant strains in the expression system to form recombinant strains from strain M/P-1-S32U to strain M/P-6-S32U (Table 1).

### 3.3. Cultivation of the Recombinant Strains

For shake flask cultivation, 20 μL of glycerol stock strain was inoculated into 20 mL LB medium (supplemented with 100 mg L^−1^ ampicillin or 50 mg L^−1^ kanamycin) and then was cultivated at 37 °C in a rotary shaker at 220 rpm for 12 h. 500 μL of above seed culture was inoculated in 25 mL terrific broth (TB) medium (supplemented with 100 mg L^−1^ ampicillin or 50 mg L^−1^ kanamycin) in a 100 mL shake flask and then was cultivated at 37 °C and 220 rpm. When the optical density at 600 nm reached 0.6–0.8, 0.01 mM isopropyl-β-d-thiogalactoside (IPTG) was added to induce expression of the recombinant proteins [26]. After that, the culture temperature was reduced to 25 °C, and it was cultivated at 220 rpm for another 24 h [25].

For fermentation in a 10 L fermenter, 10% (*v*/*v*) concentration of inoculum was inoculated into 4 L of fermentation medium for cultivation at 37 °C. The synthetic medium used for batch fermentation contained 10.0 g L^−1^ glycerol, 1.7 g L^−1^ MgSO_4_·7H_2_O, 1.0 g L^−1^ ammonium citrate, 2.0 g L^−1^ NaCl, 4.0 g L^−1^ (NH_4_)_2_HPO_4_, 1.0 g L^−1^ (NH_4_)_2_SO_4_, 6.0 g L^−1^ KH_2_PO_4_, 3.0 g L^−1^ K_2_HPO_4_·H_2_O and 10.0 mL L^−1^ trace metal solution. The trace mental solution contained 5.0 g L^−1^ CaCl_2_, 1.3 g L^−1^ sodium acetate, 0.25 g L^−1^ CoCl_2_, 1.25 g L^−1^ EDTA, 1.5 g L^−1^ MnCl_2_, 0.18 g L^−1^ ZnSO_4_, 3.0 g L^−1^ CuSO_4_, 10.0 g L^−1^ FeSO_4_, 0.3 g L^−1^ Na_2_B_4_O_7_·10H_2_O and 20.0 mL L^−1^ phosphoric acid. When the dissolved oxygen (DO) starts to go up, glucose feeding with an initial rate of 5 g/L/h was started, in which constantly adjust the feeding rate to keep the specific growth rate of cells at 0.25 h^−1^ and the concentration of glucose was less than 5 g/L [27]. Oxygen concentration was kept at 30% by controlling the impeller speed (300–650 rpm) and the air flow rate (0.167–0.833 vvm). The pH was kept at 7.0 by addition of 25% NH_4_OH or lactic acid when required. The yield rate of the target protein was calculated by means as the average production in per liter per hour throughout the fermentation.

### 3.4. Preparation of the Recombinant Proteins and SDS-PAGE

The preparation of the recombinant enzymes was performed as our previous report [25]. The cultivation broth was treated with ultrasonication at 50 Hz for 5 min in ice water. The disrupted mixture was centrifuged at 13,800× *g* for 5 min. The supernatant was the soluble protein fraction and the disrupted pellet was the intracellular denatured protein fraction as inclusion body. The recombinant proteins in each fraction were determined by sodium dodecyl sulfate polyacrylamide gel electrophoresis (SDS-PAGE) under denatured conditions. Electrophoresis was performed with a 5% stacking gel and a 12% separating gel. Protein bands were visualized by staining with Coomassie Brilliant Blue R-250 dye. Content of target protein on the gel of SDS-PAGE was estimated by software BandScan5.0.

### 3.5. Active Assay of Glycosyltransferase by HPLC

The glycosyltransferase activity of the recombinant UGT76G1 was evaluated by the conversion rate from the substrate to the goal product. The reaction mixture with a total volume of 200 μL contained 40 μL soluble glycosyltransferase samples, 20 μL of rebaudioside A (100 mM) or rebaudioside D (20 mM), 20 μL of UDP-glucose (60 mM), and 120 μL of sodium phosphate buffer (50 mM, pH 7.0). Reactions were performed at 35 °C with 220 rpm shaking for 3–24 h. The reaction was terminated by adding 160 μL of methanol solution (60%, *v*/*v*) and 16 μL of 2 N sulfuric acid solution. The mixed solution was then centrifuged at 13,800× *g* for 10 min and the supernatant was carried out by membrane filtration for product analysis by high performance liquid chromatography (HPLC). The steviol glycosides (SGs) concentration was determined using an Innoval C18 ODS-2 column (250 mm × 4.6 mm, Bonna-Agela Technologies, Tianjin, China) maintained at 40 °C with UV detection at 210 nm in a Waters e2695 HPLC system. The mobile phase was consistent of 68% acetonitrile and 32% water and the flow rate was 0.5 mL/min, the injection volume was 10 µL. St (50%, mixed with 50% RebA), Reb A (50%, mixed with 50% St) and Reb D standard (97.6%) for HPLC was purchased from Shanghai PureOne Biotechnology Co., Ltd. (Shanghai, China). The conversion rate (%) indicated the yield of target products which were calculated as follows:Conversion rate of St (%) = (A1 (Reb A))/(A0 (St)) × (805/967) × 100(1)
Conversion rate of Reb D (%) = (At (Reb M))/(A0 (Reb D)) × 1129/1291) × 100(2)
where A0 (St) and A0 (Reb D) represent the initial concentration of the substrates St and Reb D respectively, and A1 (Reb A) and At (Reb M) respectively represent the increased concentration of terminal products Reb A and Reb M after reaction.

### 3.6. Enzymatic Biotransformation of SGs Using Recombinant UGT76G1 In Vitro

For in vitro enzymatic synthesis of Reb A or Reb M by the recombinant UGT76G1, the UDP-glucose regeneration reacting system was performed as previous reports [10,28] with minor modification. The reaction solutions (20 mL) mixing St with the final concentration of 10 mM or Reb D with the final concentration of 2 mM, 80 mM (final concentration) of sucrose, recombinant UGT76G1 0.4 g/L (final concentration), and SUS1 0.2 g/L (final concentration) in sodium phosphate buffer (50 mM, pH 7.0) were reacted in 50 mL flasks at 35 °C for 3–12 h with shaking at 220 rpm. During reaction, the samples were removed at defined time points, and their reaction was terminated by adding 160 μL of methanol solution (60%, *v*/*v*) and 16 μL of 2 N sulfuric acid solution. They were then centrifuged at 13,800× *g* for 10 min and the supernatant was immediately analyzed by HPLC according to above-mentioned method.

## 4. Conclusions

In the current work, we successively used the fusion partner, co-expression of *prpD* and *malK* simultaneously, and fed-batch fermentation to increase the heterologous expression of glycosyltransferase encoding gene *UGT76G1* in *E. coli*. The final yield of the soluble recombinant enzyme Smt3-UGT76G1 was up to 1.97 g/L which showed a well scale production potential. Its application in biotransformation of SGs showed high yield of Reb A (4.8 g/L) from St and Reb M (1.8 g/L) from Reb D respectively. The high expression level of the recombinant glycosyltransferase Smt3-UGT76G1 in *E. coli* would make the low cost of production of SGs and promote the development of sweeteners industry. Besides, this article revealed a novel strategy for the enhanced expression of heterologous proteins in *E. coli*.

## 5. Patents

The research results about the construction of novel efficient *E. coli* expression system in this work has been applied for a Chinese invention patent. The patent name was “An enhanced bio-robust *Escherichia coli* chassis cell and its construction method and applications”. The patent acceptance number was 202010661070.7. 

## Figures and Tables

**Figure 1 ijms-21-05752-f001:**
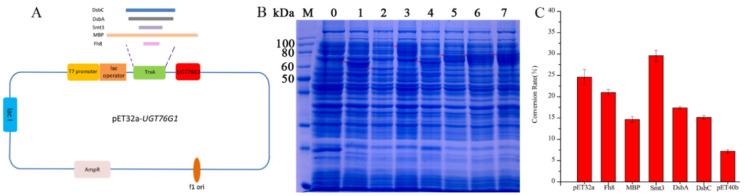
Expression of UGT76G1 with different fusion partners. (**A**): Construction diagram for fusion expression plasmids; (**B**): SDS-PAGE for different recombinant fusion enzymes in the whole cell crushing fluids. M: proteins standard markers; 0: the whole cell crushing fluid of *E. coli* BL21 (pET32a); 1: expression of TrxA-UGT76G1; 2: expression of Fh8-UGT76G1; 3: expression of MBP-UGT76G1; 4: expression of Smt3-UGT76G1; 5: expression of DsbA-UGT76G1; 6: expression of DsbC-UGT76G1; 7: expression of DsbC(pET40b)-UGT76G1; (**C**): the different conversion rates from St to Reb A in 3 h reaction by different recombinant enzymes. Data in panel C are presented as mean ± SD (*n* = 3).

**Figure 2 ijms-21-05752-f002:**
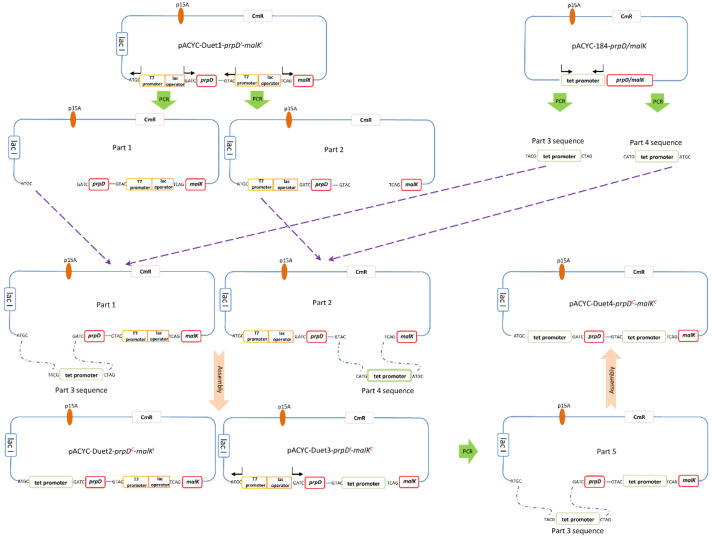
Diagram for the construction of plasmids in the overexpression system *E. coli* BL21 (DE3) M/P-(1–6). pACYC Duet1 was the original plasmid which harbouring two identical inducible promoters (T7 promoter and lac operator); pACYC Duet2 substituted the first inducible promter of pACYC Duet1 by a constitutive promoter (tet promoter); pACYC Duet3 substituted the second inducible promter of pACYC Duet1 by a constitutive promoter (tet promoter); pACYC Duet4 substituted both the inducible promters of pACYC Duet1 by constitutive promoters (tet promoter).

**Figure 3 ijms-21-05752-f003:**
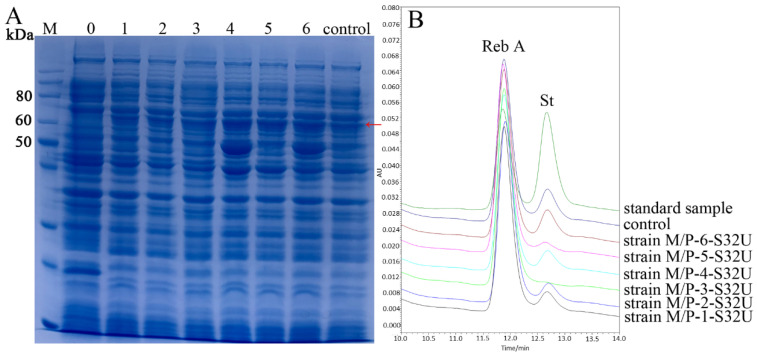
Overexpression of Smt3-UGT76G1 in the overexpression system *E. coli* BL21 (DE3) M/P-(1-6). (**A**): SDS-PAGE of the total soluble proteins of recombinant strains in expression system *E. coli* BL21 (DE3) M/P-(1–6)-S32U. M means protein marker; 0 means the total soluble proteins of *E. coli* BL21 (DE3) which harbouring plasmid pET32-Smt3; 1–6 means the total soluble proteins of recombinant strains in the expression system, respectively; control means the total soluble proteins of strain S32U; all of the total soluble proteins were diluted twice to load on the SDS-PAGE; The red arrow indicates the target protein (Smt3-UGT76G1). (**B**): The testing results of HPLC of the transglycosylation products from St by the coarse enzymes of recombinant strains in expression system. standard sample means the substrates which harbouring the same amount of pure St and Reb A; control means the transglycosylation products by the coarse enzyme of strain S32U; strain M/P-1-S32U~strain M/P-6-S32U means the transglycosylation products by the coarse enzymes of recombinant strains in expression system.

**Figure 4 ijms-21-05752-f004:**
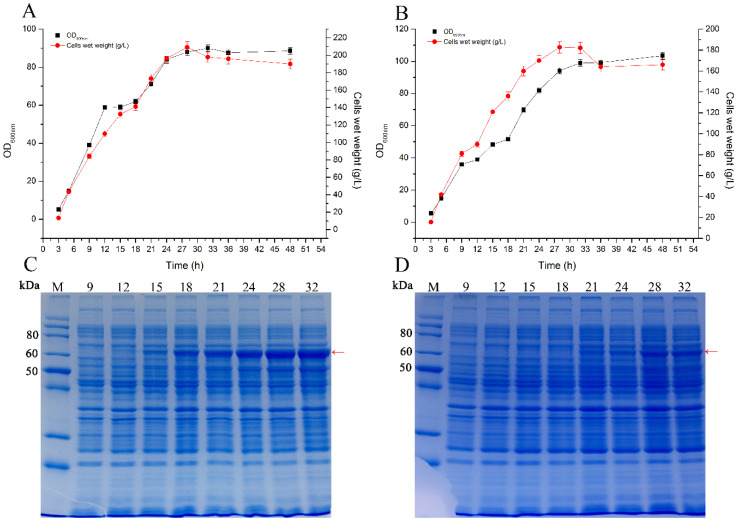
Scaling up fermentation of the recombinant strains in a 10 L fermenter. (**A**): the growth profile of stain M/P-3-S32U; (**B**): the growth profile of strain S32U; (**C**): SDS-PAGE of the soluble proteins produced by strain M/P-3-S32U at different fermentation times; (**D**): SDS-PAGE of the soluble proteins produced by strain S32U in different fermentation times; The red arrows indicate the target proteins (Smt3-UGT76G1); Data presented in panels B and C as mean ± SD (*n* = 3).

**Figure 5 ijms-21-05752-f005:**
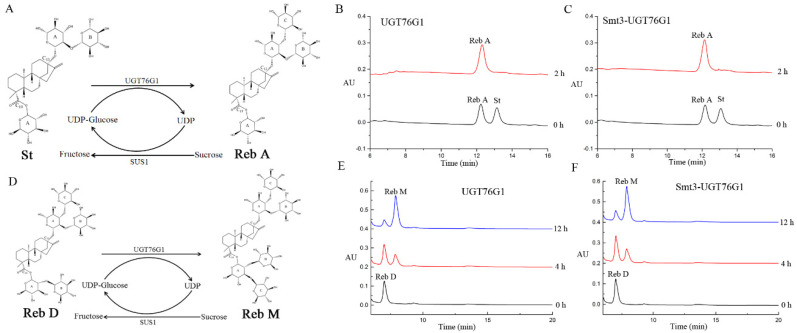
Transglycosylation reactions by the recombinant UGT76G1 in vitro enzymatic catalysis. (**A**): reaction formula of transglycosylation converted St to Reb A in the UDP-glucose regeneration transglycosylation system; (**B**): transglycosylation results converted St to Reb A catalyzed by UGT76G1 (deleted fusion partner) in different reaction times; (**C**): transglycosylation results converted St to Reb A catalyzed by Smt3-UGT76G1 in different reaction times; (**D**): reaction formula of transglycosylation converted Reb D to Reb M in the UDP-glucose regeneration transglycosylation system; (**E**): transglycosylation results converted Reb D to Reb M catalyzed by UGT76G1 (deleted fusion partner) in different reaction times; (F): transglycosylation results converted Reb D to Reb M catalyzed by Smt3-UGT76G1 in different reaction times.

**Table 1 ijms-21-05752-t001:** Strains and plasmids in the efficient expression system.

Strains	Harboured Plasmids
*E. coli* BL21 (DE3)	
Strain M/P-1	pACYC184-*malK*
Strain M/P-2	pACYC184-*prpD*
Strain M/P-3	pACYCDuet1-*prpD^I^*-*malK^I^*
Strain M/P-4	pACYCDuet2-*prpD^C^*-*malK^I^*
Strain M/P-5	pACYCDuet3-*prpD^I^*-*malK^C^*
Strain M/P-6	pACYCDuet4-*prpD^C^*-*malK^C^*
Strain M/P-1-S32U	pACYC184-*malK*; pET-Smt3-32-*UGT76G1*
Strain M/P-2-S32U	pACYC184-*prpD*; pET-Smt3-32-*UGT76G1*
Strain M/P-3-S32U	pACYCDuet1-*prpD^I^*-*malK^I^*; pET-*Smt3*-32-*UGT76G1*
Strain M/P-4-S32U	pACYCDuet2-*prpD^C^*-*malK^I^*; pET-*Smt3*-32-*UGT76G1*
Strain M/P-5-S32U	pACYCDuet3-*prpD^I^*-*malK^C^*; pET-*Smt3*-32-*UGT76G1*
Strain M/P-6-S32U	pACYCDuet4-*prpD^C^*-*malK^C^*; pET-*Smt3*-32-*UGT76G1*
Strain S32U	pET-*Smt3*-32-UGT76G1

*^I^* indicated that the gene was under inducible promoter (see Figure 2); *^C^* indicated that the gene was under constituent promoter (see Figure 2).

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
