# Peer review of "Enhanced Heterologous Production of Glycosyltransferase UGT76G1 by Co-Expression of Endogenous prpD and malK in Escherichia coli and Its Transglycosylation Application in Production of Rebaudioside"

_ijms, 2020, doi:10.3390/ijms21165752_

Round 1

Reviewer 1 Report

In this work, authors constructed an E. coli overexpression system for otherwise low-yielded glycosyltransferase UGT76G1 by fusion tags and co-expression of endogenous prpD and malK. The potential industrial application of the enhanced expression of UGT76G1 is compelling in enzymatic production of steviol glycosides Reb D and Reb M, low-calorie sugar alternatives which are scarce in their natural source.

Experiments are well designed and the clarity of results is simply exemplary. Acceptance for publication is recommended.

Reviewer 2 Report

The manuscript by Shu et al. reports the set-up of an efficient heterologous E. coli expression system for the production of a Stevia rebaudiana UDP-glycosyl transferase, based on a co-expression strategy involving previously identified helper genes (malK/prpD). In addition, a scaling-up procedure using a fed-batch fermentation, was also proposed for the most efficient system identified. The results appear relevant for biotechnological application in the sweeteners industry, but also for the co-expression methodology used. Although the methodological approach is well defined and the results are sufficiently well presented, the following items has to be faced prior the consideration for its publication on IJMS.

  1. Use the term "expression system" instead of "strain" throughout the manuscript, when describing a specific heterologous expression system set-up.
  2. How the production of 1.97 g/l of Smt3-UGT76G1 and its yield rate were assessed ?
  3. In Table 1, the expression system (strain) called E. coli BL21 (DE3) included also the transfromation with the empty pACYC184 plasmid? If so, please indicate it in the Table.
  4. In the SDS-PAGE images in Figs. 1, 3, and 4, indicate the size of the protein molecular weight standards used and use a simple lane identification (with numbers or letters) with their description in the caption.
  5. Higher dimension for the characters text in all figures should be used.
  6. To render the text more fluent, in the name of the expression systems identified avoid the repetition "E. coli BL21 (DE3)" (either in the text and in the Table 1), as it has been used for all of them.

Furthermore, a significant English language editing is required, as the text in some paragraphs looks like a spoken rather than written English. Hereafter a non comprehensive list of correction is proposed.

  • Line 112, add “in panel C” after “Data”.
  • Line 177, change “Compare” with “Compared”.
  • Lines 187 and 188, change “in” with “at”.
  • Line 189, delete "are"; add "in panels B and C" after “presented”.
  • Line 191, change "previous reports" with "previously reported", and "has the specific ....... glucose"" with "catalyses the transfer of a glucose moiety"
  • Line 192, change "A" with "residue".
  • Line 193, change "catalysis" with "catalyse"; add a comma after "Reb M".
  • Line 195, delete "respectively".
  • Line 196, add ", respectively" after "(Figure S3)"; delete "the"; change "reaction" with "reactions".
  • Line 201, change "conclude" with "be concluded".
  • Line 203, change "catalysis" with "catalytic".
  • Line 205, move "position" after "C19".
  • Line 206, change ". It is consistent" with "consistently".
  • Line 224, delete "which".
  • Line 232-233, change "could be" with "were".
  • Line 240, change the first "the" with "specific" and "took" with "and".
  • Line 252, change "Transform the" with "Transformation of the E. coli BL21 (DE3) with the".
  • Line 254, change "to E. coli ..... to form a" with "lead to the formation of".
  • Line 255, change "(Table 1)" with "listed in Table 1".
  • Line 258, change "Strains" with "Expression systems"; delete "åM/P (1-6)"
  • Line 264, delete "Inoculated"; change "at" with "was inoculated in".
  • Line 266, please control the concentration of IPTG reported; usually higher concentrations are used (0.4 mM) for induction.
  • Line 267, change "were" with "was".
  • Line 275, delete "85%".
  • Line 320, change "successively using" with "successfully used the"
